# Lignin-Degrading Bacteria in Paper Mill Sludge

**DOI:** 10.3390/microorganisms11051168

**Published:** 2023-04-29

**Authors:** Magda Rodriguez-Yupanqui, Magaly De La Cruz-Noriega, Claudio Quiñones, Nélida Milly Otiniano, Medardo A. Quezada-Alvarez, Walter Rojas-Villacorta, Gino A. Vergara-Medina, Frank R. León-Vargas, Haniel Solís-Muñoz, Segundo Rojas-Flores

**Affiliations:** 1Escuela de Ingeniería Ambiental, Facultad de Ingeniería y Arquitectura, Universidad Cesar Vallejo, Trujillo 13007, Peru; mrodriguez@ucv.edu.pe; 2Instituto de Investigación en Ciencia y Tecnología, Universidad César Vallejo, Trujillo 13001, Peru; 3Laboratorio de Biotecnología e Ingeniería Genética, Departamento de Microbiología y Parasitología, Universidad Nacional de Trujillo, Trujillo 13011, Peru; 4Departamento de Ingeniería Ambiental, Universidad Nacional de Trujillo, Trujillo 13011, Peru; 5Programa de Investigación Formativa, Universidad Cesar Vallejo, Trujillo 13001, Peru; 6Facultad de Ingeniería Civil y Ambiental, Universidad Nacional Toribio Rodríguez de Mendoza de Amazonas, Chachapoyas 01001, Peru; 7Departamento de Ingeniería Química, Facultad de Ingeniería Química, Universidad Nacional de la Amazonia Peruana, Iquitos 16002, Peru; 8Escuela de Ingeniería Industrial, Facultad de Ingeniería, Universidad Cesar Vallejo, Trujillo 13007, Peru; 9Vicerrectorado de Investigación, Universidad Autónoma del Perú, Lima 15842, Peru

**Keywords:** paper mill sludge, black liquor, laccase activity, *Agrobacterium tumefasciens*, *Klebsiella grimontii*, *Beijeinckia fluminensis*

## Abstract

The effluents generated in the paper industry, such as black liquor, have a high content of lignin and other toxic components; however, they represent a source of lignin-degrading bacteria with biotechnological potential. Therefore, the present study aimed to isolate and identify lignin-degrading bacteria species in paper mill sludge. A primary isolation was carried out from samples of sludge present in environments around a paper company located in the province of Ascope (Peru). Bacteria selection was made by the degradation of Lignin Kraft as the only carbon source in a solid medium. Finally, the laccase activity (Um-L^−1^) of each selected bacteria was determined by oxidation of 2,2′-azinobis-(3-etilbencenotiazolina-6-sulfonate) (ABTS). Bacterial species with laccase activity were identified by molecular biology techniques. Seven species of bacteria with laccase activity and the ability to degrade lignin were identified. The bacteria *Agrobacterium tumefasciens* (2), *Klebsiella grimontii* (1), and *Beijeinckia fluminensis* (1) were reported for first time. *K. grimowntii* and *B. fluminensis* presented the highest laccase activity, with values of 0.319 ± 0.005 UmL^−1^ and 0.329 ± 0.004 UmL^−1^, respectively. In conclusion, paper mill sludge may represent a source of lignin-degrading bacteria with laccase activity, and they could have potential biotechnological applications.

## 1. Introduction

The paper industry generates large volumes of black liquor as by-products and liquid waste from papermaking processes. In these processes, the pulpwood is transformed into paper pulp by removing components such as lignin, hemicellulose, and other materials to release cellulose fibers that are part of black liquor [1]. Black liquor is a complex mixture of compounds that contains a high amount of lignin (35–45%), cellulose, hemicellulose, tannic acid, resin acids, plastids, chlorinated phenol, chlorinated hydrocarbon, surfactant and biocides, acetic acid, formic acid, saccharine acids, carboxylic acids, and hundreds of other components, being, therefore, an extremely complex mixture. Likewise, these have a high total dissolved solids content, leading to a high chemical oxygen demand (COD) and turbidity [2,3,4,5,6].

Black liquor is a difficult effluent to treat due to its complex composition, presenting economic limitations, low-efficiency physicochemical methods, and bioremediation strategies with intrinsic difficulties caused by high pH (10–13) and lignin content [7]. In the paper industry, black liquor undergoes primary and secondary biotreatment processes for its degradation. However, during the secondary treatment process, some biopolymers, such as cellulose, hemicellulose, lignin, and chlorolignin, are degraded to some extent, but some compounds are only biotransformed as the metabolic product [8]. Kraft lignin (KL) is the main pollutant in effluents generated by the paper industry. The complex structure of KL represents a challenge for the scientific community due to its resistance to the depolymerization process [9]. On the other hand, the direct discharge of KL into water (for example, the ocean) generates a negative impact, putting aquatic life and the food chain at risk [3].

One of the main challenges in the paper production industry is the elimination of lignin. These heteropolymer and chlorinated phenols are the major persistent organic pollutants in pulp and paper mill effluents, contributing to color and toxicity [10,11]. Lignin is an irregular, randomly crosslinked phenolic polymer composed of three different phenylpropane monomers connected by C–C bonds and ether bonds. Due to this structure complex, the degradation speed is slow under natural conditions [12]. Over the years physical/chemical procedures have been implemented for the treatment of lignin However, these processes are usually expensive and dangerous for the environment due to the generation of effluents with highly toxic substances. On the other hand, biological treatments use microorganisms or enzymes to degrade lignin. These enzymes (phenol oxidases and heme peroxidases) are called ligninolytic and catalyze the oxidative cleavage of lignin. The biochemical characteristics of these enzymes give them great potential in industrial processes [10].

The major lignin-degrading enzymes are laccase (Lac), lignin peroxidase (LiP), manganese peroxidase (MnP), and versatile peroxidase (VP) [13,14]. Laccase is a multi-cooper oxidase in various microorganisms [14]. This enzyme is classified as benzenediol oxygen reductase (EC 1.10.3.2) and it can oxidize many phenolic and nonphenolic molecules due to its low substrate specificity, using oxygen as an electron acceptor and generating water and by-products [15]. Laccases, LiP, and MnP are the ligninolytic (hemoprotein) enzymes involved in the oxidation of many aromatic compounds. Meanwhile, VP, also known as hybrid peroxidase (manganese-lignin peroxidase), contains glycoproteins and belongs to the oxidoreductase family [13,15,16].

In previous studies, some bacteria were isolated from pulp and paper mill wastewater, for example, *Bacillus aryabhattai* [17], *Bacillus subtilis*, *Micrococcus luteus*, and *Phanerochaete crysosporium* [18]. Another study shows a ligninolytic strain of *Klebsiella*, which presented genes involved in the degradation of lignin [19]. Recently, Elframawy et al. [20] studied extracellular peroxidases from bacteria belonging to the *Streptomycetaceae* family, which degrade various substrates, such as guaiacol, vanillic and veratric acids, and alkaline lignin. Similarly, Granja-Travez et al. [21] established that some bacteria exhibit diverse metabolic strategies used for lignin oxidation, while the β-ketoadipate pathway appears to be the most common pathway for aromatic metabolism in lignin-degrading bacteria.

Some studies suggested that lignin degradation bacteria have a biotechnological potential to be used in bioremediation. Singh et al. [9] studied the biodegradation of KL by *B. aryabhattai*, which can be used in the treatment of other lignin-containing industrial effluents. Similarly, Kumar et al. [22] investigated the effective degradation of KL from paper effluent and identified *Bacillus cereus* degrading KL from activated sludge.

In this sense, the objective of the research was to isolate and identify species of lignin-degrading bacteria from sludge formed by effluents discharged into the soils surrounding the paper company located in Ascope. Bacteria with the ability to degrade lignin could be used in future research to assess lignin residues.

## 2. Materials and Methods

### 2.1. Sample Collection and Physical-Chemical Characterization

The samples were taken from sludge formed by the mixture of effluents from a paper company with the soil adjacent to the company. The sampling area was located in the province of Ascope (Peru) (Figure 1). Three sludge samples were collected. The co-ordinates of the sampling area are 7°57′57.6″ S 79°14′56.4″ W, 7°57′58.2″ S 79°14′56.3″ W, and 7°57′56.7″ S 79°14′52.7″ W. Samples of 500 mL volume were transferred to the Clean and Emerging Technologies Laboratory at Universidad Nacional de Trujillo (Peru) and then stored at 4 °C until their use.

For the characterization of the sample, a physical-chemical analysis was carried out. The color (PCU, Platinum Cobalt Units) was described according to the Munsell system, while pH and conductivity were determined from the supernatant using a multiparameter instrument (Orion Stan^TM^ A329; Thermo Scientific; Waltham, MA, USA). Turbidity was assessed through the nephelometric method of Standard Methods. For another part, lignin, cellulose, and hemicellulose content were measured according to Jönsson et al. [23].

### 2.2. Isolation and Selection of Lignin-Degrading Bacteria

#### Primary Isolation and Selection

Three samples were mixed (1.5 L) and a representative 10 mL sample was immediately chosen. A dilution (1:10 *v*/*v*) with saline solution (0.85%) was performed from a representative sample. This dilution was then stirred in a vortex and allowed to stand for 5 min until the supernatant had formed. Serial dilutions (10^6^) of the supernatant were made. From the last dilution, it was seeded on nutrient agar. These culture media were incubated at 30 °C for 24 h. Subsequently, a pure culture of each isolated bacterial colony was carried out.

The capability to degrade lignin was proof for each isolated bacterium. For the selection of lignin-degrading bacteria, a culture medium with lignin as the only carbon source was employed. The minimum salt medium (MSM) contained (g/L) Na_2_HPO_4_.2H_2_O, 7.8; KH_2_PO_4_, 6.8; MgSO_4_, 0.2; NaNO_3_, 0.085; NH_4_ (CH_3_COO) Fe, 0.01; Agar powder, 20.0; and alkaline lignin, 5.0. The culture media were incubated at 30 °C for 48 h. Finally, the colonies with different morphology were selected and axenic (pure) culture was performed. These were stored at 4 °C [24].

### 2.3. Laccase Activity of Selected Bacteria

#### 2.3.1. Enrichment for Enzyme Production

The medium used for crude enzyme production was Berg salt liquid medium, composed of 2 g NaNO_3_; 0.5 g K_2_HPO_4_; 0.02 g MgSO_4_:7H_2_O; 0.02 g MnSO4:7H2O; 0.02 g FeSO_4_:7H_2_O; 0.5 g CaCl_2_:2H_2_O; and 1% alkaline lignin (Sigma Aldrich) in 1L of distilled water [25]. To obtain the crude enzyme from each bacterium, it was necessary to make suspensions, incubated at 30 °C and shaken at 120 RPM for 15 days. Then, they were centrifuged at 7000 rpm for 6 min and the supernatant was separated for enzymatic evaluation.

#### 2.3.2. Laccase Activity Test

The activity of lacasse (Lac) was tested by oxidation of nonphenolic dye 2,2′-azinobis-(3-ethylbenzethioline-6-sulfonate) (ABTS) using the method developed by Lai et al. [26]. This enzyme mixture consists of 0.15 mL of 0.03% ABTS, 0.5 mL of 0.1 M sodium acetate buffer at pH 5.0, and 0.35 mL of sample supernatant. The formation of oxidized ABTS was measured in a quartz cuvette of 1 mL (light passage of 1 cm) and a wavelength of 530 nm using a UV-Vis spectrophotometer. The activities were expressed in UmL^−1^. A single unit of enzyme activity equates to the oxidation of 1 micromole of lignin per minute under the test conditions, utilizing the molar extinction coefficient. The oxidation of ABTS was followed by an absorbance increase at 420 nm with a molar extinction coefficient of ε420 = 32,000 M^−1^ cm^−1^. This activity was evaluated in triplicate.
Laccase activity (UmL^−1^) = ((Abs × 1000000U)/32000 × 0.350 mL)/1000 (1)

### 2.4. Molecular Identification and Phylogenetic Analysis of the Isolates

For molecular identification, an axenic culture of each bacterium was sent to Ecobiotech Lab SAC Laboratory (Lima, Peru). The procedure described by this laboratory was the following: DNA bacterial extraction was performed using the Cetyl Trimethyl Ammonium Bromide (CTAB) method. Then, the 16 rRNA gene was amplified by PCR technique using 27F and 1492R primers. Sanger sequencing was used to sequence the amplicons by Macrogen Inc. (Seoul, South Korea). The Bioinformatic Software MEGA 11 (Molecular Evolutionary Genetics Analysis) analyzed the sequences. Finally, they were aligned and compared with other sequences in the bioinformatic program Basic Local Alignment Search Tool (BLAST), by which the percentage of identity for the identification of bacteria species will be obtained.

The phylogenetic tree was performed using the methodologies of Maximum Likelihood (ML) and Bayesian Inference (BI) in RAxML v.7.2.6 [27] and MrBayes v.3.2.5 [28], respectively. In RAxML, the GTR + I + Γ replacement model was used for the amplified region (16S rDNA) and the option of fast Bootstrap with 1000 replications. Meanwhile, for MrBayes, the GTR + I+G evolutionary model based on the Akaike Information Criterion (AIC) was first determined in ModelTest v.2.2.10 [29]. Finally, BI was performed in two analyses consisting of 4 MCMC chains for 1,100,000 generations, with one sampling every 200 generations, and 10% of trees (burn-in) were discarded.

### 2.5. Data Analysis

The statistical analysis of the values of the laccase activity was carried out using SPSS software version 22.0. The values of the means were used, which were obtained from experiments in triplicate. Anova Unifactorial analysis was applied to evaluate significant differences between the means obtained of enzymatic activity for each type of bacteria, with an alpha significance of 5%. For analysis of multiple comparisons, a post hoc analysis with Tukey’s HSD test was performed.

## 3. Results

### 3.1. Physical-Chemical Characterization of the Liquid Sample

The characterization of the sludge sample is shown in Table 1. The characteristics show evidence of soil contamination with effluents with black liquor. The color is related to the lignin concentration because this heteropolymer is responsible for the brown color. The sample’s pH alkaline (8.75 ± 0.017) is characteristic of paper industry effluents and gives an idea of what type of microorganisms grows in this type of sample.

### 3.2. Isolation and Selection of Lignin-Degrading Bacteria

A total of 11 isolated bacteria were obtained from the sample, coded as B-1 to B-11. All the isolated bacteria showed different morphology in Nutritive Agar. However, only seven isolates were selected because they were the only ones that grew from lignin as the only carbon source in the medium (B 1–7).

### 3.3. Laccase Activity

The colorimetric method (qualitative) showed evidence of laccase activity of the seven selected bacterial isolates. Brown coloration indicates oxidation of 2,2′-azinobis-(3-ethylbenzethioline-6-sulfonate) (ABTS) substrate.

Concerning Table 2, different values of laccase activity were reported at pH 5.0. Regarding the values of laccase activity, strain B-7 produced the highest activity (0.329 ± 0.004d UmL^−1^), while the lowest value was recorded in strain B-1 (0.181 ± 0.003a UmL^−1^).

Table 2 also presents a summary of the Tukey HSD test, which shows a significant difference (*p* < 0.05) between the enzyme activity produced by the seven selected bacteria. Four internally homogeneous groups were generated but different from each other, where strains B-1 (group 1) and B-3 (group 2) differ significantly from the rest; however, in strains B-2, B-4, and B-5 (group 3), equality of means was found (*p* = 0.27). In the same way, for strains B-6 and B-7 (group 4), the probability was *p* = 0.18, with a 95% confidence interval. However, B-6 and B-7 have high enzymatic activity (UmL^−1^) in comparison to the other strains.

### 3.4. Molecular Identification of Lignin-Degrading Bacteria

The BLAST analysis of the 16S rRNA gene sequence of the isolated bacteria showed the closest phylogenetic relationships with other species, as shown in the phylogenetic tree (Figure 2).

On the other hand, the selected bacteria, B-1; B-2; B-3; B-4; B-5; B-6; and B-7, were identified by the 16S rRNA gene (Table 3). The species of identified bacteria were *Agrobacterium tumefaciens* (2), *Agrobacterium* sp. (2), *Bacillus* sp. (1), *Klebsiella grimontii* (1), and *Beijeinckia fluminensis* (1).

## 4. Discussion

According to physical-chemical characteristics of the sample, soil contamination by effluents with black liquor is evident. In addition, the characteristics of black liquor depend on the conditions of wood processing [26]. The dark brown color is given by the concentration of lignin which is released in the process of lignocellulosic degradation; the greater the lignin, the greater the color intensity of the black liquid [30]. The pH of the sample was alkaline (8.75 ± 0.17); however, it can vary from 7 to more than 13 [31,32,33]. This variation is due to the pH of the white liquor used in the early phases of cooking during processing to obtain cellulose. White liquor has a pH of 13–14 and is rich in hydroxide ions [34]. Turbidity and total solids are determined by organic components that come from black liquor and the environment and the presence of microbial communities [3,35,36]. Finally, the lignin content (84.04 ± 3.70 g/L) comes from the black liquor produced in the paper industry [1]. The recalcitrance of lignin due to its complex structure [37,38] will allow selective isolation of lignolytic microorganisms.

Seven colonies of selected bacteria had the ability to degrade lignin, which is a complex process involving different lignolytic enzymes (LiP, MnP, PV, and Lac) [23,39,40,41,42,43]. The activity of the laccase (Table 2) was determined by the oxidation of the ABTS substrate (brown coloration) and evidences the enzymatic machinery of the bacteria isolated from paper sludge [44]. One of these enzymes is the laccase, which acts in synergy with other peroxidases, providing better percentages of lignin degradation [45]. Recent advances in the mechanisms of lignin degradation by laccase have been studied, finding that there is formation of hydrogen bonds and the participation of certain residues of conserved amino acids (His419, Cys492, His497, and Met502) [46,47]. On the other hand, the laccase activity values obtained in the seven isolates are higher than those reported (2.28 UL^−1^) in another study [48] and lower than the values reported (4.43 UmL^−1^) by *Bacillus paramycoides* BL2 [49]. The different values of the laccase activity of the isolates can be influenced by microbial biodiversity, which regulates enzyme production according to the existing substrate [50].

Table 3 shows that most species are in the phylum *Proteobacteria*. The laccase is frequently found in prokaryotes, where the largest number of strains with genes expressed by the laccase belong to phylum *Proteobacteria*, *Actinobacteria*, and *Fimicutes*, while in *Cyanobacteria* and *Bacteroides* the number of strains with these genes is lower [51]. Regarding identified microorganisms (Table 3), it was possible by sequencing the 16S rRNA gene. It is the first report of some lignin-degrading bacteria strains (*Agrobacterium tumefasciens*, *Klebsiella grimontii*, and *Beijeinckia fluminensis*).

On the other hand, Figure 2 shows the phylogenetic relation between seven bacteria isolated capable of lignin degradation. The species of *Agrobacterium* have been reported as lignin-degrading [52]. Regarding the enzymes that these species can express, Si et al. (2015) characterized a new gene (*atm*) that expresses a laccase of *Agrobacterium* sp. strain S5-1, which improved the digestion of corn stover [53]. On the other hand, a combination of peroxidase enzymes of the DyP type with the accessory enzyme Agro LigE enhances the degradation of polymeric lignin, releasing low-molecular-weight by-products [54]. Meanwhile, another study by Faisal et al. [55] sequenced the genome of *Agrobacterium* sp. strain S2, identifying various genes that express enzymes with the ability to degrade aromatic compounds derived from lignin, such as superoxide dismutase, cytochrome P450, catalase-peroxidase, non-heme chloroperoxidase and benzaldehyde dehydrogenase, and other aromatic ring oxidizing genes, such as 4-hydroxybenzoate polyprenyltransferase and 4-hydroxyphenylacetate 3-monooxygenase. These findings allow these bacteria to be used for the pretreatment of residues based on lignin and lignocellulose [56].

Regarding the species of *Bacillus* sp. BL.4, it was possible to isolate it from the black liquor sample. Different species of *Bacillus* have been reported in samples from effluents from paper manufacturing companies [57], as well as *Bacillus altitudinis* SL7 [58], *Bacillus aryabhattai* [9,18], *Bacillus paramycoides* strain BL2 [49], *Bacillus cereus* [23], among others [59]. These species can proliferate in black liquor at an alkaline pH (8.75 ± 0.17) due to the enzymatic machinery degrading lignin as peroxidases and laccase [9,23,58,60]. Within the genome of *Bacillus* genes can be found encoding laccase, cytochrome P450, and vanillin alcohol oxidase, as well as 64 genes involved in multiple aromatic metabolic pathways, such as phenylalanine metabolism and aminobenzoate degradation, as discovered in *B. cereus* AH7-7 [61].

As for the species of *Klebsiella grimontii* B-6, this is a species reported for the first time in this study. *Klebsiella* spp. species are widely distributed in natural environments, such as soils, waters, plants, etc. [62]; then, they can be found in samples involving the use of cellulose-based raw materials, such as black liquor. In this way, these species of bacteria may produce enzymes capable of degrading lignin as *Klebsiella pneumoniae* NX-1 [63]. Genomic studies in *Klebsiella* sp. BRL6-2 strain can encode four peroxidases, including glutathione peroxidase and DyP-type peroxidases [64]. Meanwhile, the genome of *Klebsiella variicola* P1CD1 has 262 genes associated with lignin-modifying enzymes and lignin degradation helper enzymes necessary for the degradation of lignin and aromatic compounds [65].

Finally, the species *Beijerinckia fluminensis* B-7 has not been reported so far as a lignin degrader. However, the genus *Beijerinckia* is distributed in the soil and the phyllosphere of plants [66], so it could be in contact with lignin residues and express some of the enzymes involved in its degradation, as demonstrated with the laccase activity determined in this study (0.329 UmL^−1^). Similarly, it has been shown that some species of this genus can degrade aromatic hydrocarbon molecules [67], demonstrating that this species may be able to degrade complex structures, such as lignin. Finally, Figure 2 shows that *B. fluminensis* B-7 appears to be related to the genus *Agrobacterium*, as is mentioned in other studies [68,69,70].

## 5. Conclusions

It is possible to find lignin-degrading bacteria in sludge samples that are formed in soils by contamination with paper mill effluents. Lignin-degrading bacteria such as *Agrobacterium tumefaciens*, *Klebsiella grimontii*, and *Beijerinckia fluminensis* were first reported. However, future genomic characterization studies of the newly isolated species are required to identify the genes involved in lignin degradation and understand the metabolic pathways involved. On the other hand, through these species, it is possible to give added value to waste with high lignin content, such as to treatment of effluents generated in the paper manufacturing industry.

## Figures and Tables

**Figure 1 microorganisms-11-01168-f001:**
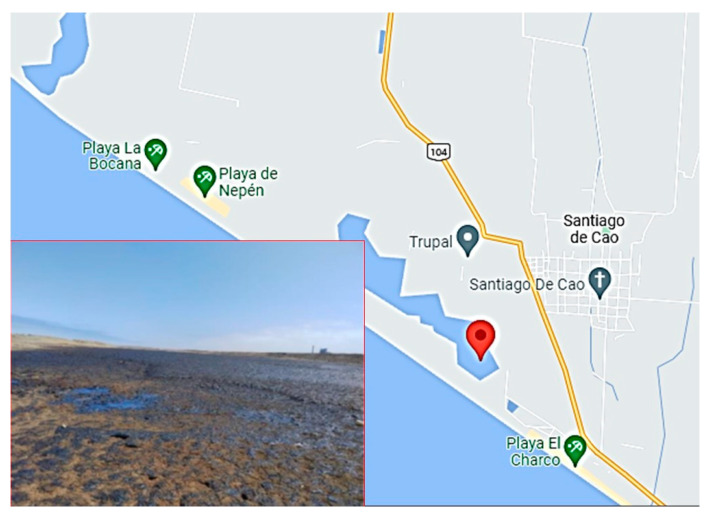
Location of the sampling area in the province of Ascope (Peru).

**Figure 2 microorganisms-11-01168-f002:**
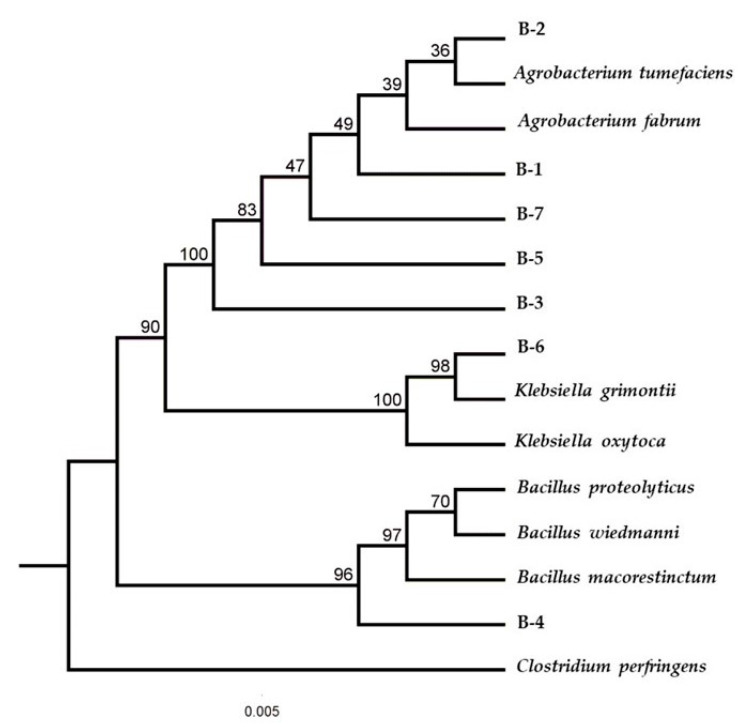
Phylogenetic tree of isolated and selected lignin-degrading bacteria with their related neighboring species.

**Table 1 microorganisms-11-01168-t001:** Physical-chemical characterization of the sample.

Parameters	Result
Color (PCU)	498,300 ± 16,237
pH	8.75 ± 0.17
Conductivity (mS/cm)	41.18 ± 2.73
Turbidity (NTU)	31,000 ± 1587
Total Solids (g/L)	140.84 ± 14.28
Lignin (g/L)	84.04 ± 3.70
Cellulose	-
Hemicellulose	-

**Table 2 microorganisms-11-01168-t002:** Laccase activity of lignin-degrading bacteria.

Sample ID	Value ± SD (UmL^−1^) *
B-1	0.181 ± 0.003a
B-2	0.292 ± 0.005b
B-3	0.275 ± 0.005c
B-4	0.294 ± 0.005b
B-5	0.301 ± 0.005b
B-6	0.319 ± 0.005d
B-7	0.329 ± 0.004d

* SD: standard deviation obtained in three repetitions. The different letters show significant differences.

**Table 3 microorganisms-11-01168-t003:** Identified species using 16S rRNA gene sequencing from samples.

Sample ID	Phylum	Identified Species	% Identity	Access Number
B-1	*Proteobacteria*	*Agrobacterium tumefaciens*	99.09	CP040640.1
B-2	*Proteobacteria*	*Agrobacterium tumefaciens*	98.58	CP040641.1
B-3	*Proteobacteria*	*Agrobacterium* sp.	84.81	NR_041396.1
B-4	*Firmicutes*	*Bacillus* sp.	88.71	NR_115526.1
B-5	*Proteobacteria*	*Agrobacterium* sp.	93.26	NR_041396.1
B-6	*Proteobacteria*	*Klebsiella grimontii*	99.72	NR_159317.1
B-7	*Proteobacteria*	*Beijerinckia fluminensis*	98.86	NR_116306.1

## Data Availability

Not applicable.

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
