# Peer review of "Lignin-Degrading Bacteria in Paper Mill Sludge"

_microorganisms, 2023, doi:10.3390/microorganisms11051168_

Round 1

Reviewer 1 Report (Previous Reviewer 2)

Dear Authors

The manuscript sent to me for review entitled "Paper mill effluents: A source of lignin-degrading bacteria with biotechnological potential" contains the results of an interesting study on the isolation and identification of lignin-degrading bacteria from paper mill effluents. Compared to the previous version of the manuscript, the current version has been improved and supplemented with missing methodological elements. 

My comment concerns the discussion section. I propose to remove references to Tables 1, 2 and 3 in L-263, 319 and 342. Such references should be included in the results section. 

Author Response

Dear colleague, thank you very much for the comments made.
The response to the comments made is sent.

* My comment concerns the discussion section. I propose to delete references to Tables 1, 2 and 3 in L-263, 319 and 342. Such references should be included in the results section.

Ans. The above changes were considered.
Best regards

Reviewer 2 Report (New Reviewer)

In this work, authors isolated several bacteria from the soil site, contaminated with paper mill eddluents. Some of these bacteria were shown to have laccase activity which implies their potential biotechnological significance. This should be the scope of future work. The manuscript has a good Introduction, however I have questions about other sections:

1. The title should be changed to the more relevant assuming that:

-the bacteria were isolated from the mixture of effluent and soil, meaning that the bacteria could have come from the soil and not from the effluent

- in your work, you did not show biotechnological potential. What has been done is the isolation and testing of laccase activity.

2. Knowing that "The major lignin-degrading enzymes are laccase (Lac), lignin peroxidase (LiP), manganese peroxidase (MnP), and versatile peroxidase (VP)", why did you only test laccase activity?

3. Line 35: "Likewise, the selected species showed laccase activity." Why do you repeat this thesis? See lines 31-32

4. Please revise Lines 64-65, 71-72, 73-74, 81, 98, 116, 130, 131, 132-133, 148, 206, 226. You have too many punctuation errors, incorrect English, grammatical errors.

5. Lines 105-106: "effluents, which present biotechnological potential." biotechnological potential is not shown on real waste.

6.  Lines 122-123: "For the characterization of the sample, a physical-chemical analysis was carried out. Parameters such as pH, conductivity, turbidity, and color (spectrophotometric technique)" Please give a more efficient description of the methods used.

7.  Lines 136-138: "The minimum salt medium (MSM) contained (g/L): Na2HPO4.2H2O, 7.8; KH2PO4, 6.8; MgSO4, 0.2; NaNO3, 0.085; NH4 (CH3COO) Fe, 0.01; Agar powder, 20.0 and alkaline lignin, 5.0" 

Could CH3COO be a source of carbon? Also, to date, many microorganisms have been identified that can hydrolyze and metabolize agar as a source of carbon and energy. Could it be that some of the bacteria could consume the agar and not the lignin?

8. Lines 146-147: "26]. Approximately 7 colonies of pure bacteria have been inoculated..."

Strange expression... Could not you somehow more accurately count the number of colonies added?

9. Something wrong with Equation 1. mL divided by mL cannot give mL-1 units.

10. Lines 199-201: "However, only 7 isolates were selected because they were the only ones that grew from lignin as the only carbon source in the medium (B 1-7)."

What did the rest of the bacteria grow on then?

11. Table 2, Column "Oxidation of ABTS"

This column is useless, since from the next column it is already clear that ABTS is oxidized, isn't it?

12. Lines 232: "On the other hand, the selected bacteria (1,2,3,4,5,6,7)" Please use the same bacteria designations, B-1, B-2, etc.

13. Lines 234-236: "These species are common environmental habitats and show the diversity of microorganisms found in black liquor." 

Please rephrase. Strictly speaking, the sample was not black liquor, but a mixture of black liquor and soil.

14. Lines 242-243: "Table 1 shows the sample's composition; according to its physical-chemical characteristics, it could be black liquor"

If there is any doubt that this is black liquor, then everywhere in the text you need a prefix "possibly" or "presumably"

15. Lines 242-260: This text is not needed in the Discussion. This does not apply to the topic of work on the isolation of ligninolytic bacteria and testing laccase activity.

16. Lines 268-280: here you are discussing what kind of method it is, which has been used for a long time and there is no point in discussing it. You modified it, what's new in that?

17. Line 336: "..related to the genera Agrobacterium.." 

not genera, but genus

the genus name should be in italics

18. Abstract and Conclusion should be modified according to above comments.

Author Response

Dear colleague, thank you very much for comments made, the authors made the changes and answered each question.
I hope you like it.

1. The title should be changed to the most relevant one on the assumption that:
The bacteria were isolated from the effluent-soil mixture, meaning the bacteria could have come from the soil and not from the effluent
-In his work by him, he showed no biotechnological potential. What has been done is the isolation and testing of the laccase activity.
Ans. The title was rewritten based on the observations, leaving the following title: Lignin-degrading bacteria in soil polluted with paper mill sludge, which better reflects what was done in the investigation.
2. Knowing that "the main enzymes that degrade lignin are laccase (Lac), lignin peroxidase (LiP), manganese peroxidase (MnP) and versatile peroxidase (VP)", why did you only analyze the activity of laccase?
Answer: The investigation of LiP, MnP and VP was a limitation of our investigation, however, the laccase activity and the mere fact of growing in a medium where only lignin (kraft lignin) is the only carbon source, evidences the ability of isolates to degrade lignin.
3. Line 35: "Likewise, the selected species showed case activity." Why do you repeat this thesis? See lines 31-32
Answer: That error has been corrected.
4. Review lines 64-65, 71-72, 73-74, 81, 98, 116, 130, 131, 132-133, 148, 206, 226. It has too many punctuation errors, incorrect English, grammatical errors.
Answer: Corrected the suggestion.
5. Lines 105-106: "effluents, which present biotechnological potential". The biotechnological potential is not shown in the actual waste.
Answer: Corrected the suggestion.
6. Lines 122-123: "For the characterization of the sample, a physical-chemical analysis was performed. Parameters such as pH, conductivity, turbidity and color (spectrophotometric technique)" Please describe more efficiently the methods used.
Ans.  The methods and equipment used were included.
7. Lines 136-138: "The minimum saline medium (MSM) contained (g/L): Na2HPO4.2H2O, 7.8; KH2PO4, 6.8; MgSO4, 0.2; NaNO3, 0.085; NH4 (CH3COO) Fe, 0.01; Agar powder, 20.0 and alkaline lignin, 5.0"
Can CH3COO be a carbon source? In addition, to date, many microorganisms have been identified that can hydrolyze and metabolize agar as a source of carbon and energy. Could it be that some of the bacteria could consume agar and not lignin?
Answer:
-Although acetate can be metabolized by microorganisms, the amount that is in the medium is insufficient, in such a way that it does not act as a source of carbon and energy.
-On the other hand, some microorganisms with the ability to degrade agar have been documented, however, agar as a gel is resistant to microbial degradation (https://doi.org/10.1016/j.btre.2019.e00346) and the enzyme capable of degrading agar has a slight effect on agar gels (https://doi.org/10.1099/00221287-87-1-150). Finally, agar-degrading enzymes are mostly produced by marine gram-negative bacteria (https://doi.org/10.3389/fmicb.2020.01934).
-Since they are bacteria that have already been in contact with lignolithic compounds, they are capable of degrading them before agar, since it is a compound with which they have not been in contact, and they would also need an adaptation phase when the first carbon source is exhausted. (Kraft lignin), which would be unlikely, since by the time they were allowed to grow they would only be metabolizing Kraft Lignin and not agar.
8. Lines 146-147: "26]. Approximately 7 colonies of pure bacteria have been inoculated..."
Strange expression... Couldn't you somehow count more accurately the number of colonies added?
Answer: the error was corrected, the sentence was rewritten.
9. There is something wrong with Equation 1. mL divided by mL cannot give mL-1 units.
Answer: The equation was corrected, there was a writing error.
10. Lines 199-201: "However, only 7 isolates were selected because they were the only ones that grew from lignin as the only carbon source in the medium (B 1-7)."
What did the rest of the bacteria grow up on?
Answer. Only seven colonies of bacteria were selected, because of the 11 colonies, only seven bacteria had the ability to degrade lignin as the only carbon source. The medium used as detailed in the procedure had only Kraft salts and lignin as the sole carbon source.
11. Table 2, Column "Oxidation of ABTS"
This column is useless, since the next column already shows that the ABTS is rusty, right?
Answer: Table 2 was corrected.
12. Lines 232: "On the other hand, the selected bacteria (1,2,3,4,5,6,7)" Use the same designations of bacteria, B-1, B-2, etc.
Answer: Fixed.
13. Lines 234-236: "These species are common environmental habitats and show the diversity of microorganisms found in black liquor."
Please reformulate. Strictly speaking, the sample was not black liquor, but a mixture of black liquor and earth.
Answer: the sentence was removed.
14. Lines 242-243: "Table 1 shows the composition of the sample, according to its physicochemical characteristics it could be black liquor"
If there is any doubt that this is black liquor, everywhere in the text you need a prefix "possibly" or "presumably".
Answer: Fixed. The sentence was rephrased stating that the soil was contaminated by black liquor. Obviously it is most likely because the company is located a few meters from the sampling site.
15. Lines 242-260: This text is not necessary in the Discussion. This does not apply to the topic of work on the isolation of ligninolytic bacteria and laccase activity testing.
Answer: lines 242-260 were rewritten.
16. Lines 268-280: Here you are discussing what kind of method it is, which has been used for a long time, and there is no point in discussing it. You modified it, what's new in that?
Answer:Yes, it was modified.
17. Line 336: ".. related to the genera Agrobacterium.."
not genders, but genders
The genus name must be italicized
Answer: Fixed.
18. The summary and conclusion should be modified in accordance with the above comments.
Answer: summary and conclusions were corrected.

Best regards

Round 2

Reviewer 2 Report (New Reviewer)

Thank you very much for the work done, the manuscript looks much better now.

1. In the revised manuscript, please change title to “Lignin-degrading bacteria in soil polluted with paper mill sludge” as was proposed by you in response to my comments

2. I repeat my question. In Section “2.2.1. Primary isolation and selection” you used lignin as the only carbon source in the media, which resulted in obtaining of 11 isolates.

While, in Lines 221-223 you wrote: "However, only 7 isolates were selected because they were the only ones that grew from lignin as the only carbon source in the medium (B 1-7)." What do you mean? All the 11 isolates were obtained using lignin-based media, however only 7 isolates grew from lignin as the only carbon source?

Please address the remaining issues, and the manuscript may be published.

Author Response

Dear colleague, thank you very much for your comments; I am going to clarify the doubt, I hope it can be understood.

Answer:
For the primary isolation, it was planted in Nutrient Agar and the selection was made in minimal salts agar with Kraft lignin as the only carbon source. For this, lines 160-163 were redrafted, for their better understanding.
Lines 160-163: From the last dilution, it was seeded on nutrient agar. These culture media were incubated at 30°C for 24 hours. Subsequently, a pure culture of each isolated bacterial colony carries out.
The ability to degrade lignin was proof for each isolated bacterium.

Kind regards

This manuscript is a resubmission of an earlier submission. The following is a list of the peer review reports and author responses from that submission.

Round 1

Reviewer 1 Report

The present study aimed to identify species of lignin-degrading bacteria and determine their laccase activity by ABTS. The results showed that seven species of bacteria with lignin biodegrading potential were identified and showed laccase activity. The significance of this research is not outstanding, and the innovation is weak. The method of identifying the new species in title is also inaccurate. Detailed phylogenetic identification and functional analysis of isolated bacteria using genomics is recommended.

Author Response

Dear colleague, I hope you are very well.
The group of researchers agree with your comment in the phylogenetics part; but it was not taken into account in the research process. For future research on the subject, it will be taken into account.

Reviewer 2 Report

Dear Authors,

The paper entitled „New species of lignin-degrading bacteria from industrial effluents
discharged on the soil”
aimed to identify species of lignin-degrading bacteria and determine their laccase activity. Authors isolated and selected of lignin-degrading bacteria collected from contaminated soils near a paper manufacturing company. For the selection a culture media with lignin such as only carbon source was employed. Next the activity of Lacasse was tested. Molecular identification of the isolates was performed by sequencing the 16S rRNA gene. In results Authors identified seven species of bacteria with lignin biodegrading potential. The obtained results are interesting from cognitive and practical point of view. Isolated new species can be used in different fields of science and industry.

The manuscript is good written, but it needs major revisions.

Abstract should be supplemented with specific results on laccase activity.

L33 keyword: “lignin-degrading bacteria” occurs in the title and should be replaced with another word.

Introduction is too long, contains a lot of detailed information and results from other studies. Therefore, it should be shortened.

L 128. Please provide detailed information about the location from which the samples were taken, e.g. map, geographic coordinates. Next, how many samples were taken?

L 171. Please provide more information on which sequencing method was used, which primers were used for sequencing rRNA gene.

Please explain, why cellulase activity was not checked.

Why the activity of selected strains was not checked on other lignocellulosic waste, e.g. straw.

Citations of literature positions in the text should be corrected according to the journal's guidelines.

Author Response

Dear colleague, I hope you are very well.
Ans.Corrected according to suggestions.
Ans. Regarding the study of cellulases of the isolates, it was not carried out because our objective was only to study those bacteria that produce ligninases, mainly laccase, which is of industrial importance.
A and regarding the verification of the lignocellulosic activity of the selected bacteria will be considered in subsequent studies, since this preliminary work was only based on a primary screening of bacteria with the potential to degrade lignin.

best regards

Author Response

Dear colleague, I hope you are very well.
• Fixed according to suggestions.

best regards

Round 2

Reviewer 1 Report

The authors' responses are too perfunctory, and the revised manuscript did not see substantial improvement.

Reviewer 2 Report

The manuscript has been revised by the Authors in accordance with my comments. I make no further comments.